# The Role of Posttranslational Modifications During Ebola Virus Infection

**DOI:** 10.3390/v17121640

**Published:** 2025-12-18

**Authors:** Joaquin Moreno-Contreras, Yoatzin Peñaflor-Tellez, Ricardo Rajsbaum

**Affiliations:** Center for Virus-Host-Innate-Immunity, RBHS Institute for Infectious and Inflammatory Diseases, and Department of Medicine, New Jersey Medical School, Rutgers University, Newark, NJ 07103, USA; joaquin.moreno@rutgers.edu (J.M.-C.); yoatzin.pt@rutgers.edu (Y.P.-T.)

**Keywords:** orthoebolaviruses, post-translational modifications, replication cycle, innate immune system, ubiquitination, phosphorylation

## Abstract

Orthoebolaviruses (OEV) are highly pathogenic viruses responsible for the Ebola virus disease (EVD). To establish a successful infection, OEV hijacks the host cell machinery, which in turn responds to infection by activating cellular antiviral pathways. These processes are regulated via post-translational modifications (PTMs) of both cellular and viral proteins. The most common PTMs include phosphorylation, ubiquitination, acetylation, methylation, and glycosylation. These modifications regulate stability, activity, and interactions between proteins that control the immune response, cell metabolism, and cell death, among others. PTMs are critical during the viral replication cycle as they can be either proviral, facilitating adequate virus replication inside the infected cell, or antiviral, most commonly hindering essential viral processes such as viral genome transcription or replication. Here, we review the different roles of PTMs known to occur during OEV infection in both viral and cellular proteins. Understanding how OEV modulates the fate of host cell proteins through specific PTMs can provide a basis for the development of novel therapeutic strategies.

## 1. Introduction

The *Filoviridae* family is filamentous, enveloped, single-stranded negative-sense RNA viruses that are generally highly pathogenic and can cause severe disease. This family includes the genera *Orthoebolavirus* and *Orthomarburgvirus* [1]. The genus *Orthoebolavirus* comprises six species: *Orthoebolavirus zairense* (Ebola virus, EBOV), *Orthoebolavirus sudanense* (Sudan virus, SUDV), *Orthoebolavirus bundibugyoense* (Bundibugyo virus, BDBV), *Orthoebolavirus taiense* (Taï Forest virus, TAFV), *Orthoebolavirus bombaliense* (Bombali virus, BOMV), and *Orthoebolavirus restonense* (Reston virus, RESTV). In addition, *Orthomarburgvirus marburgense* includes Marburg virus (MARV) and Ravn virus (RAVV) [2]. The most common outbreaks in Africa have been caused by EBOV, SUDV, and BDBV. In contrast, only one human infection with TAFV has been reported [3]. Unlike the highly pathogenic EBOV, RESTV is the only member of the *Orthoebolavirus* genus that does not cause disease in humans, although it can infect and cause disease in nonhuman primates [4]. Most of the studies to date have been conducted using EBOV proteins or infectious virus; therefore, we focus our review primarily on this virus.

EBOV was discovered in 1976 during an outbreak in the Democratic Republic of Congo (DRC), near the Ebola River, from which the disease obtains its name [5]. Since then, several outbreaks of orthoebolaviruses (OEV) have been recorded in Africa, with the largest outbreak occurring in West Africa between 2013 and 2016 that resulted in over 28,000 infections and more than 11,000 deaths [6,7]. Multiple outbreaks of related highly lethal ebolaviruses have occurred in past years, raising concerns that outbreaks could spread beyond Africa due to travel and the potential of causing a lethal pandemic (Table 1). The most recent outbreaks this year include the related SUDV in February 2025 [8], and as recent as September 2025, an outbreak of EBOV was declared in the DRC [9]. OEV spreads through direct contact with blood and other bodily fluids, as well as from tissues or contaminated fomites from an infected individual. The disease caused by this virus is known as Ebola virus disease (EVD) or Ebola hemorrhagic fever, which includes flu-like symptoms, as well as diarrhea, vomiting, impaired kidney function, and both internal and external bleeding in humans and nonhuman primates [10]. Infections with OEV are highly lethal, with human case fatality rates ranging from 40 to 90%.

### 1.1. OEV Structure and Life Cycle

The OEV virion is composed of seven structural proteins encoded by the 19-kilobase (kb)-long genomic RNA. These structural proteins include the nucleoprotein (NP), the polymerase cofactor VP35, which is also involved in type-I Interferon (IFN-I) antagonism, the matrix protein VP40, the transcriptional activator VP30; a minor matrix protein VP24, which plays a role in the nucleocapsid condensation; the envelope glycoprotein GP, which recognizes the cellular receptor in the host cell; and the large polymerase protein L, which is main component of the viral RNA-dependent RNA polymerase [12].

The primary target cells for OEV infection are mononuclear phagocytes (macrophages, Kupffer cells, and microglia) and dendritic cells (DCs) [13], but it can productively infect other cells, including epithelial and endothelial cells, and can also enter T cells, resulting in non-productive replication [14]. Infected DCs can migrate to regional lymph nodes, where the virus can further disseminate to other susceptible cell types [15]. Subsequently, these infected cells can disseminate the virus to the kidney and liver, facilitating further virus spread and resulting in high viremia [16].

Viral entry occurs through receptor-mediated endocytosis and macropinocytosis [17] triggered by the viral GP interaction with cellular surface proteins, and uses the intracellular Niemann-Pick C1 protein (NPC1) as the main entry factor [18,19,20,21]. The viral particle escapes the late endosome upon proteolytic processing of viral GP, which induces membrane–membrane fusion and nucleoprotein complex release into the cytoplasm [22,23]. The nucleoprotein complex, formed by the viral proteins NP, VP35, VP30, and L, binds to the viral genome for either the synthesis of the antigenomic positive-strand RNA, which serves as a template for genomic RNA during replication, or the synthesis of viral mRNAs that are 5′-capped and 3′-polyadenylated [24,25]. While replication of the viral genome is dependent on NP, VP35, and L, transcription of viral genes requires VP30 [26]. Viral protein expression hijacks cellular membranes and proteins to form cytoplasmic replication factories, where viral genome replication occurs [27]. At late time points after infection, structural proteins traffic through the anterograde protein secretion pathway, where they accumulate on the cell membrane. Finally, viral genome–protein complexes are formed, packaged into lipid-enveloped virions, and released through budding as the cell undergoes apoptosis [28]. The steps that regulate the shift from virus RNA replication to viral RNA transcription are still not well understood but appear to require multiple host factors and post-translational modifications (PTMs) of viral proteins, which will be discussed in detail in this review.

### 1.2. Innate Immunity to OEV

The innate immune system is the first line of defense against microbial infections. In the cytoplasm, viral infections are detected by cellular pattern recognition receptors (PRRs), which recognize pathogen-associated molecular patterns (PAMPs) [29,30]. The recognition of PAMPs triggers a signaling cascade that culminates in the expression and secretion of IFNs, which in turn induce the expression of IFN-stimulated genes (ISGs) that promote a cellular antiviral state [29]. Viral RNA can be recognized by the cytoplasmic retinoic acid-inducible gene I (RIG-I)-like receptors (RLRs), RIG-I, and melanoma differentiation-associated gene 5 (MDA5) [31,32]. After their activation, these receptors are recruited to the mitochondria via the adaptor protein (MAVS/IPS-1), which in turn promotes the activation of TANK-binding kinase (TBK1)/I-kappa B kinase e (IKKε) kinase complexes [33]. The activation of TBK1 and IKKε leads to the phosphorylation of IFN-regulation factor 3 (IRF3) and/or IRF7 [34]. Phosphorylated IRF3 and IRF7 undergo dimerization and translocation to the nucleus to promote the transcription of IFN-I genes [35,36]. Secreted IFNs then induce an antiviral state in both the infected cell and neighboring cells through the JAK/STAT pathway, which results in the expression of multiple ISGs and other immune signaling molecules [37,38]. However, activation of these innate immune pathways also triggers the induction of multiple inflammatory cytokines, which, in excess, can cause tissue damage. This dysregulation of the innate immune response is one of the hallmarks of EVD. EBOV is highly efficient in blocking the antiviral IFN-I response via the viral VP35 protein, which sequesters double-stranded RNA (dsRNA) from PRRs [39].

## 2. Post-Translational Modifications (PTMs) During Virus Infection

To avoid hyper-inflammation, innate immune signaling is tightly regulated by post-translational modifications (PTMs) [40]. PTMs constitute a dynamic and reversible mechanism to modulate protein activity, viability, expression, conformation, localization, interactome, and stability. These modifications occur through the covalent or non-covalent attachment of specific chemical groups to amino acid side chains, either enzymatically or non-enzymatically [41,42]. The most commonly studied PTMs include phosphorylation, ubiquitination, acetylation, glycosylation, methylation, and SUMOylation [43]. Recently, other modifications such as myristylation, palmitoylation, prenylation, and sulfation have also been proven to modulate several pathways [44]. Moreover, protein can be modified by multiple PTMs, either cooperatively or competitively. Viral infection can induce either activation or inhibition of enzymes involved in the addition or removal of PTMs to cellular or viral proteins to aid in viral replication [45,46,47,48]. Likewise, the cellular immune response can use PTMs to induce viral protein inactivation, degradation, or modulate cellular protein activity to impede or slow down virus cycle progression.

OEV has developed a variety of strategies to overcome recognition and activation of the antiviral response mediated by IFN-I [49,50]. These strategies include inhibiting the activation of proteins of innate immunity pathways, often through PTMs on cellular and viral proteins, thereby avoiding recognition of viral PAMPs by cytoplasmatic PRRs [51,52]. This review focuses on PTMs that occur on viral and host proteins during OEV infection and how these modifications influence OEV replication and pathogenesis, as well as the antiviral immune response of the host cell.

### 2.1. Phosphorylation

Protein phosphorylation is one of the most studied post-translational modifications. It plays a critical role in regulating protein–protein interactions, protein stability, intracellular localization, gene transcription, signal transduction, and cell cycle regulation [53]. This reversible modification is carried out by protein kinases. These enzymes transfer the γ-phosphate group from ATP to the hydroxyl group of the target amino acid residue.

Protein kinases are among the largest gene families in eukaryotes, and the human genome encodes over 500 kinases, representing 1.7% of the entire genome [54]. It has been estimated that around one-third of proteins encoded in the human genome are phosphorylated at some point during their life cycle, with an estimated 100,000 different phosphorylation sites in the cellular proteome [55,56]. Most of the phosphorylation events occur on serine, threonine, and tyrosine residues. Among these, phosphorylation at serine residues is the most abundant, representing 86.4%, followed by threonine with 11.8%, and finally the tyrosine residue with only 1.8%. In rare cases, the phosphorylation of histidine and aspartate residues also occurs [57].

Phosphorylation is a reversible PTM, in which the phosphate group can be removed from phosphoproteins by the action of phosphatases. There are approximately 226 known protein phosphatases [58]. Since phosphorylation acts as a molecular switch for many regulatory proteins, dysregulation of kinase activity is associated with several human diseases. The most common mechanism by which phosphorylation exerts its function is via protein–protein interactions that depend on phospho-binding domains present in partner proteins. For example, SH2 domains are commonly known to bind phospho-tyrosines on target proteins [59,60].

#### 2.1.1. Phosphorylation During OEV Replication

The viral replication cycle is well known to require phosphorylation and dephosphorylation of both cellular and viral proteins, enabling the regulation of a wide range of cellular processes such as intracellular transport, apoptosis, and immune response, among others. Phosphorylation of viral proteins is involved in the regulation of viral gene expression, nucleocapsid nucleation, viral genome replication, morphogenesis, and egress from infected cells. In general, viruses rely on host cellular kinases, with the exception of herpesvirus [61], which encode their own protein kinases. OEV, such as other viruses, hijacks host kinases to either enhance replication or to block phosphorylation-dependent immune signaling. More importantly, some of the dysregulation of the immune response that causes EVD is due to aberrant expression or dysregulated activity of host kinases/phosphatases, leading to abnormal cytokine/chemokine expression.

##### VP30 Phosphorylation

Ebola virus VP30 principal function is to act as a viral RNA transcription initiation factor. VP30 activity is tightly regulated by a reversible phosphorylation cycle. The N-terminal of VP30 contains two serine clusters at positions 29–31 and 42–46, as well as a threonine residue at position 52 (T52), which are located near its RNA-binding domain [62]. The phosphorylation of six serine residues in these clusters affects VP30s ability to bind RNAs and regulates its interaction with VP35 and NP, modulating the composition of the viral polymerase complex, inhibiting viral transcription, and favoring replication [63]. It has been suggested that phosphorylation of VP30 leads to its dissociation from the VP35-L complex, while favoring its interaction with NP [63]. The serine–arginine protein kinase 1 (SRPK1) has been identified as the cellular kinase responsible for the phosphorylation of residue S29 in VP30, which is located within the N-terminal R_26_xxS_29_ motif. The interaction between VP30 via the R_26_xxS_29_ motif recruits SRPK1 into EBOV-induced inclusion bodies, and inhibition of endogenous SRPK1 impairs the primary viral transcription (Figure 1A) [64]. Whether other cellular kinases are involved in modulating the phosphorylation of VP30 remains to be evaluated.

It has been proposed that protein phosphatase 1 (PP1) is a host cell protein involved in regulating the phosphorylation status of VP30 by controlling its dephosphorylation [65]. However, recent studies have shown that the host phosphatase PP2A-B56 can also mediate VP30 dephosphorylation. The interaction between PP2A-B56 and VP30 is mediated by the viral nucleoprotein (NP). NP contains two motifs, LxxIxE and PPxPxY, which are recognized by PP2A-B56 and VP30, respectively. These motifs are in close proximity and are required for the dephosphorylation of VP30 and the modulation of viral transcription [66]. Interestingly, both motifs are conserved across all filovirus NP proteins, suggesting that this is a conserved mechanism to regulate the switch between replication and transcription of the viral genome (Figure 1B). Whether PP1 can directly dephosphorylate VP30 and the specific relevance of this protein in the context of viral infection remains to be evaluated.

Additionally, in an in vitro system, it was found that VP30 is also phosphorylated at T143 and T146, which promote VP30 phosphorylation within the regions containing serine clusters at residues 29–31 and 42–46, located close to the RNA-binding domain. Viral particles can package phosphorylated VP30 at T146 [67], suggesting that VP30 phosphorylation may also play a function in viral RNA packaging. Overall, while it is a consensus in the field that dephosphorylated VP30 is required for initiation of viral RNA transcription, it is still unclear what makes the viral polymerase L-VP35 complex commit to function as a replicase or transcriptase.

##### VP35 Phosphorylation

The most well-known function of VP35 is as an antagonist of IFN-I induction and does so via multiple mechanisms. The C-terminal region of VP35, which contains the IFN-inhibiting domain (IID), binds and sequesters viral double-stranded RNA (dsRNA) produced during viral replication, preventing the activation of the cytoplasmic sensor RIG-I [39]. Additionally, VP35 inhibits the activation of IRF-3 by interacting with the kinases IKKε and TBK-1, competing with IRF3 [68]. While VP35 can be phosphorylated by these kinases, the interaction between VP35 and IKKε has been characterized in more detail, revealing that this interaction occurs through the amino-terminal domain of IKKε. Furthermore, VP35 disrupts the interaction between IKKε and IRF-3/IRF-7, thereby reducing IRF-3/7 phosphorylation and inhibiting IFN-α/β gene expression [39,68]. Recently, it was shown that viral inclusion bodies induced by transcription- and replication-competent ebolavirus virus-like particles (trVLPs) contribute to immune evasion by sequestering IRF-3 VP35-STING interaction, preventing IRF-3 phosphorylation and translocation into the nucleus, which is critical for IFN-I production [69]. However, it is unclear why or how STING would be involved in this process, since STING is activated upon recognition of dsDNA downstream of cGAS [70]. It could be that the involvement of STING is due to the presence of transfected DNA expression vectors, which are required for the trVLP system; therefore, this is unlikely to happen during actual EBOV infection.

VP35 is also an essential cofactor of the viral polymerase, and as such, the activity of VP35 in virus transcription/replication can also be regulated by phosphorylation. In two independent studies, specific phosphorylation sites in VP35 were evaluated using high-resolution liquid chromatography-linked tandem mass spectrometry (LC-MS/MS). This analysis identified ten phosphorylated S/T residues (S187, T191, S195, S205, T206, T207, S208, T210, S310,317, and S317). Of these amino acid residues, only five (S187, S205, T206, S208, and S317) were consistently identified in both studies. The role of these phosphorylation sites was subsequently assessed using an EBOV minigenome system, which suggested that phosphorylation on S187 and T210 is critical for the transcriptional activity of the Ebola virus polymerase complex (Figure 1B) [71]. Additionally, phosphorylation at T210 was found to be important for the interaction between the viral proteins VP35 and NP, although it did not seem to play a functional role in IFN-I antagonism [67]. Together, these studies indicate that VP35 phosphorylation is key to modulating its activity, either as a component of the polymerase complex or in its role as an IFN-I antagonist. In addition, while phosphorylation of VP35 seems to favor transcription versus viral RNA replication, and VP30 does not seem to be required for replication, other PTMs, including ubiquitination of VP35 (see more details below), could modulate the function of VP35 as a cofactor of the viral polymerase to switch from vRNA replication to RNA transcription. How PTMs on different viral proteins contribute to the regulation of transcription versus replication is still not well understood.

Although the importance of VP35 phosphorylation has been explored thoroughly, the kinases that phosphorylate VP35 have been less studied. So far, only TBK-1 and IKKε have been identified, but the function of these kinases has not been associated with direct pro-viral functions. Therefore, it is possible that additional kinases may phosphorylate VP35 to regulate its direct pro-viral function as a cofactor of the polymerase. Identifying the kinases involved in VP35 phosphorylation and developing specific inhibitors for these kinases could inform the development of novel therapeutic options for EBOV.

##### NP Phosphorylation

The main function of NP is to encapsidate the viral RNA, protecting it from cellular RNases, but it also plays important roles in virus replication, transcription, and viral RNA packaging into the virions. The N-terminal domain of NP is important for the formation of NP–NP dimers, oligomers, and NP-RNA structures, while the C-terminal is important for incorporating nucleocapsids into virions [72]. In the host cell, Marburg virus (MARV) NP exists in both phosphorylated (S/T residues) and non-phosphorylated states; however, only the phosphorylated form has been detected in virions [73,74]. Using LC-MS/MS of HEK-293 cells expressing EBOV NP, phosphorylation was detected on residues T563, S581, S587, and S647 in the C-terminus of NP. These sites are conserved on NPs from EBOV and MARV. In contrast to MARV NP, both phosphorylated and non-phosphorylated forms of EBOV NP are present in virions [75]. Recently, a new phosphorylation site on NP at position T603 (NP-T603) has been reported in EBOV virions. Interestingly, the phosphorylation status of NP at this position differs between virions and infected cells. Upon viral infection, NP-T603 is dephosphorylated to support efficient primary viral transcription (Figure 1D). Moreover, NP phosphorylation appears to modulate the VP30-dependent transcription initiation by facilitating VP30 dephosphorylation mediated by PP2A-B56α [76].

In addition to its role in VP30 phosphorylation, it was recently reported that PP1 also binds to NP. This interaction appears to be relevant for NP dimerization and EBOV capsid formation. Mechanistically, NP binding to PP1 prevents NP dimerization and facilitates NP recruitment to the EBOV transcription complex. Importantly, this interaction does not affect the NP-PP2A binding, which is important for the VP30 dephosphorylation (see previous section) [77]. Whether the PP1-NP interaction affects the phosphorylation of NP, modulating its function, remains to be determined. Further investigation is required to determine the significance of NP phosphorylation in other steps of the EBOV replication cycle, such as packaging, transport, or egress of nucleocapsid. The cellular kinases responsible for this PTM have not yet been identified.

##### GP Phosphorylation

The viral glycoprotein (GP) is a heavily glycosylated type I transmembrane protein present on the surface of virions responsible for receptor binding, fusion of the virus with the host cell, and as a virulence factor. While no evidence of functional phosphorylation on GP has been reported, it has been shown that the mucin-like domain of GP can reduce the phosphorylation of ERK1/2, with a higher effect on the ERK2 kinase isoform (Figure 1C). The inactivation of ERK1/2 enhances GP-induced toxicity [78]. It might be technically difficult to identify GP phosphopeptides (or other PTMs) by MS because of the high glycosylation status of GP.

##### VP40 Phosphorylation

The VP40 matrix protein is a peripheral membrane protein that surrounds the nucleocapsid complex, forming regular protein arrays beneath the lipid bilayer. VP40 is critical for OEV budding, nucleocapsid recruitment, virus structure, and stability [23,79]. The expression of VP40 induces VLP production in a virus-free context. In EBOV VP40, the following two phosphorylation sites have been reported: Tyrosine 13 and Serine 233, which are phosphorylated by the cellular kinases c-Abl1 and Cdk2/CycE, respectively. While phosphorylation at Y13 is important for the replication of EBOV in vitro [80], S233 appears to be relevant for its packing and release into exosomes [81]. In contrast, VP40 of MARV is phosphorylated at tyrosine residues at positions 7, 10, 13, and 19. The phosphorylation of these residues is important for the recruitment of nucleocapsids structures to budding sites and the proper assembly of infectious virus particles (Figure 1D) [82]. The identification of the kinases responsible for the phosphorylation of MARV VP40 protein, and their role in the phosphorylation of VP40 from other OEV, remains to be identified.

### 2.2. Ubiquitination

Ubiquitination is a reversible PTM in which Ubiquitin (Ub), a 76 amino acid protein, is covalently attached most commonly to a lysine (K) residue of a target/substrate protein [83]. This PTM can modulate many different cellular functions, including protein activity, localization, association with other biomolecules, and protein stability [84].

Canonical protein ubiquitination involves the sequential action of three enzymes, the Ub-activating enzyme E1 [85], an Ub-conjugating enzyme E2, and an E3-Ub ligase [86]. E3-ligases provide specificity to the reaction by recognizing the substrate and enabling the transfer of Ub from the E2 to a K residue on the target protein via a thioester bond. Mammals encode two E1 activating enzymes, over thirty E2 enzymes, and more than five hundred E3 ubiquitin ligases have been described to date [87]. Deubiquitinating enzymes can remove Ub chains from the target protein, making this translational modification reversible, and allowing the recycling of cleaved Ub chains.

Proteins can be mono, poly-mono, or poly-ubiquitinated. Since Ub itself has seven lysines, polyubiquitin chains can be linked through lysine residues K6, K11, K27, K29, K33, K48, and K63 with the carboxy terminus of another Ub chain, resulting in elongated chains. These chains have different structures (topology), which provide the ability to interact with different binding proteins, resulting in the diversification of functions [88]. K48-linked polyubiquitination has long been associated with protein turnaround and stability because these chains have a high affinity for the proteasomal subunits, leading to the targeting of K48-linked polyubiquitinated proteins for degradation. K63-linked polyubiquitination is associated with signaling modulation in several cell processes, including immune response and cancer progression, due to the presence of ubiquitin-binding domains (UBD) present in effector protein partners. Other linkages, such as K11, K27, and K29, are less well understood and remain active research topics to this date [89].

It is important to note that free Ub chains, which are not covalently anchored to any protein (also called unanchored Ub), have also been shown to play important functions in immune signaling and virus replication [49,90,91,92]. Although the function of these chains has been more controversial, it is clear that these chains can modulate protein activity in vitro. Since unanchored Ub does not act via a covalent modification of a K residue, it is usually necessary to make multiple mutations on the target binding protein to break the interactions, allowing for the study of function. Demonstrating a direct role of unanchored Ub is complicated by the lack of reagents. However, using different indirect techniques, including the deubiquitinase Isopeptidase T (IsoT, also called USP5), which specifically cleaves unanchored Ub, or using inhibitors to block Ub-substrate non-covalent interactions, there is overwhelming evidence that unanchored Ub also plays a role in cells and in vivo. Unanchored polyubiquitin chains have been detected in purified EBOV virions [93], as well as in other viruses [94,95,96], suggesting they do play a role in virus infectivity. Therefore, new innovative approaches are needed to further demonstrate the significance of unanchored Ub during infection.

#### 2.2.1. Ubiquitination During OEV Infection

Reports of viral proteins undergoing covalent ubiquitination date back to the 1970s. Protein ubiquitination can have either proviral functions by facilitating adequate virus replication inside the infected cell, or antiviral activity, by inducing viral protein degradation through the proteasome system and hindering vital viral processes such as viral genome transcription or replication, or immune evasion.

Ubiquitin-mediated activation of cellular factors involved in the antiviral immune response, as well as the degradation of host proviral factors through the proteasome pathway, and on the other hand, ubiquitination of viral factors regulating protein–protein interactions necessary for proviral functions results in a dual role of this PTM during viral infection. Like the vast majority of viruses studied to date, OEV does not encode its own Ub-ligase system [47]. Instead, it uses the host Ub-ligases to hijack this PTM and facilitate various steps of its replication cycle. It is also important to note that, such as other PTMs, ubiquitination of both viral and cellular factors occurs in a cell type-dependent manner and is also time-dependent. Therefore, expression of the Ub-ligases in specific cell types could determine virus tropism and is an area that needs more in-depth studies.

##### NP Ubiquitination

Using a transcription and replication competent virus-like particle (trVLP) system, it was recently reported that EBOV NP is ubiquitinated by the E3-Ub ligase TRIM25. It was proposed that TRIM25 acts as an antiviral factor against EBOV as part of the IFN-I response. In this model, NP ubiquitination by TRIM25 limits viral infection by promoting NP dissociation from viral RNA. Then, free viral RNA becomes accessible to the Zinc finger antiviral protein ZAP, which binds to the viral RNA in a CpG-dependent manner and targets it for degradation (Figure 2A) [97]. Although these studies propose an interesting novel antiviral role of TRIM25, the experiments were all conducted in cell lines and not with actual infectious EBOV. While the trVLP system can be useful to answer certain mechanistic questions, it is not always reliable to conclusively demonstrate what happens during infection. The trVLP system requires overexpression of different viral proteins, which do not represent biological conditions during infection. In addition to this, there is overwhelming evidence that TRIM25 promotes IFN-I induction via activation of RIG-I [98], although a small number of studies dispute this fact [99]. Therefore, the function of TRIM25, especially in overexpression systems and in cell lines, remains controversial. It remains to be seen if TRIM25 possesses antiviral function during infection with live EBOV in primary cells and in vivo.

##### VP35 Ubiquitination

EBOV VP35 is a multifunctional protein involved in viral genome transcription, replication, and IFN-I suppression, and ubiquitination plays multiple roles to modulate its activity. We have reported that TRIM6 interacts with VP35 and ubiquitinates the K309 residue present on the C-terminal IID, facilitating polymerase cofactor activity and promoting EBOV replication [100]. We also reported that recombinant EBOV encoding VP35 mutations K309R and K309G replicate less efficiently than WT EBOV in both IFN-I-competent and -deficient cells. TRIM6 ubiquitination on VP35 K309 promotes efficient viral transcription along a transcriptional gradient by stabilizing interactions with the viral RNA polymerase L and preventing interaction with NP protein, delaying viral nucleocapsid assembly (Figure 3A) [101]. The VP35-K309R mutant virus also showed increased IFN-I inhibition, suggesting that ubiquitination on VP35-K309 to enhance viral transcription and ultimately replication comes at the expense of reduced IFN-I antagonism [101]. Therefore, maintaining low levels of ubiquitinated VP35 on K309 would favor both efficient viral transcription as well as IFN-I inhibition. Since TRIM6 is also involved in the IFN-I response [91] and TRIM6 knockout cells produce significantly less IFN-I than WT cells during EBOV infection [101], this suggests that the proviral function of TRIM6-mediated ubiquitination of VP35 dominates over TRIM6 antiviral activity.

A recent study showed that the E3 ubiquitin ligase Mindbomb 2 (MIB2) binds to the EBOV VP35 NNLNS motif. MIB2 plays important roles in the regulation of innate immunity by promoting the IFN-I and NF-κB signaling pathways. The VP35–MIB2 interaction inhibits IFN-I induction, whereas a mutant lacking the NNLNS domain (VP35 5A) exhibited reduced IFN-I inhibition. Using minigenome assays, it was shown that ectopic expression of this VP35 5A mutant had reduced viral RNA polymerase activity as compared to the WT VP35. In an attempt to demonstrate that these VP35 effects are relevant in the context of infectious EBOV, a recombinant virus was generated lacking the NNLNS motif. When Vero cells (which are defective in IFN production) were infected with this VP35 5A infectious virus, no significant differences in viral titers were observed compared with the WT virus [102]. These results could imply that the function of the NNLNS motif on VP35 depends on IFN-I, which would be consistent with the role of MIB2 in the IFN pathway, but not on VP35 function as the cofactor of the polymerase. The reason why viral RNA synthesis was affected in vitro but not in infected cells remains to be determined. It was also shown that VP35 is ubiquitinated in the presence of MIB2; however, the specific VP35 lysine residue that is ubiquitinated was not identified. Whether the NNLNS motif affects other VP35 functions, such as interactions with viral proteins within the ribonucleoprotein complex, also remains to be evaluated. These further illustrate the complex interplay between VP35 ubiquitination and the multiple functions this protein performs throughout the viral replication cycle.

Additionally, we demonstrated that VP35 interacts with unanchored K63-linked Ub chains via it is IID, and that IsoT treatment that abrogates this interaction results in reduced viral replication (Figure 3B) [93]. Structural bioinformatic analysis and mutagenesis identified an important contribution of the R225 residue on VP35 in the interactions with unanchored K63-Ub chains. Interestingly, VP35 capability to bind to dsRNA [39] is enhanced when VP35 is in complex with K63-Ub chains [93]. A computational screen of FDA-approved drugs identified two molecules that could potentially break VP35–Ub interactions, pCEBS (3-[4-amino-sulfonyl) phenyl] propanoic acid) and SFC (2,5-dimethyl-4-sulfamoyl-furan-3carboxylic acid). These compounds are known to inhibit carbonic anhydrase and Metallo-β-lactamase, respectively [103,104]. These compounds inhibited interactions of Ub with VP35 in an in vitro coIP assay and significantly reduced EBOV replication in cells. Furthermore, SFC treatment significantly reduced EBOV replication and improved clinical scores in EBOV-infected mice [93]. These results not only shed light on the regulation of the multiple functions of VP35 through ubiquitination but also demonstrate the importance of unanchored Ub chains in promoting viral replication and open new approaches to fight infections by disrupting these interactions through drug repurposing.

##### GP Ubiquitination

The transmembrane OEV envelope glycoprotein (GP) is a heavily glycosylated viral protein that forms trimeric spikes on the virion surface and mediates receptor binding and membrane fusion. Initially, it is expressed as a long polypeptide called pre-GP, which is processed in the endoplasmic reticulum and Golgi apparatus and cleaved by furin, generating two disulfide-linked subunits, GP1 and GP2 (GP_1,2_). During EBOV infection, high levels of GP_1,2_ in the cytoplasm trigger cell rounding and detachment, which may contribute to its high pathogenicity [105]. To regulate these cytopathic effects during viral infection, EBOV reduces its expression of GP_1,2_ at the mRNA level, using the GP transcript to produce a nonstructural soluble GP (sGP) and a small soluble GP (ssGP) [106]. sGP plays a role in immune evasion and promotes the replication cycle by increasing the virus uptake in late endosomes [107]. A lower expression of GP_1,2_ and a higher level of sGP are optimal for EBOV replication.

One mechanism by which EBOV maintains this delicate balance is the ubiquitination of GP. GP_1,2_ is K27-linked polyubiquitinated at residue K673 by the E3-Ub ligases RNF185, TRIM25, and membrane-associated RING-CH-type 8 (MARCH8). Interestingly, although TRIM25 and MARCH8 polyubiquitinated GP_1,2_, they did not downregulate its expression. In contrast, RNF185-knockdown strongly increased GP_1,2_ expression, suggesting that RNF185-mediated ubiquitination of GP_1,2_. Surprisingly, GP_1,2_ degradation is not mediated by the proteasome but rather occurs through autophagosomes, resulting in GP_1,2_ degradation in autolysosomes [108] (Figure 2B). It has been reported that the E3-Ub ligase MARCH8 affects the GP_1,2_ processing in an E3 ligase activity-dependent manner by inhibiting N-glycan formation, O-glycosylation, and furin-mediated proteolytic cleavage of GP. This impairs GP shedding and secretion, suggesting that MARCH8 exerts antiviral activity in EBOV replication [109]. Moreover, the E3-Ub ligases MARCH1 and MARCH2, which share 64% and 20.5% homology with MARCH8, respectively, displayed a similar pattern to MARCH8 in restricting EBOV GP processing. Notably, even though MARCH2 and MARCH8 share low sequence homology, their effects on the GP processing are similar, possibly because they share a similar membrane-associated structure. MARCH1 and MARCH2 also hijacked furin-mediated proteolytic cleavage and blocked GP maturation, as observed with MARCH8 [110]. This suggests a conserved antiviral mechanism among MARCH family proteins. The cellular protein targeted by MARCH8 that mediates furin inhibition, as well as the potential role of TRIM25 in GP regulation, remains unknown.

##### VP40 Ubiquitination

The OEV matrix protein VP40 possesses a conserved proline-rich motif (PPxY) at its N-terminus, which mediates interactions with WW-domains of cellular proteins, particularly during the final stages of virion assembly and egress [111]. The interaction between the membrane-associated ubiquitin ligase Nedd4 and VP40 was demonstrated in vitro. Furthermore, a ubiquitination assay showed that Rsp5, a homolog of Nedd4 that also contains multiple WW domains, mediates VP40 ubiquitination. This ubiquitination appears to favor the efficient release of VP40 from the cell [112]. This study was performed using in vitro approaches; it is important to determine whether Nedd5 also exerts this proviral function during EBOV infection in vivo.

Similarly, the interaction between the VP40 PPxY domain and the E3-Ub ligases WWP1 and ITCH has been reported. WWP1 knockdown reduces the levels of high-molecular-weight VP40 oligomers without affecting monomer levels [113]. Although VP40 oligomerization versus monomerization was not evaluated in ITCH-knockdown assays, both cases resulted in reduced VLP budding [114] (Figure 2C). In addition to Nedd4, ITCH, and WW1P, the E3 ligase Smad Ubiquitin Regulatory Factor 2 (SMURF2) also interacts with VP40 through the binding of its WW domain to the VP40 PPxY domain. In VLP budding assays, siRNA-mediated knockdown of SMURF2 inhibits VLPs’ egress, whereas SMURF2 overexpression enhances VP40 VLP egress. Importantly, this effect depends on the enzymatic activity of SMURF2, as a catalytically inactive mutant does not affect VP40 budding [115]. Whether SMURF2 has a relevant role in EBOV egress remains to be determined. These findings suggest that VP40 ubiquitination has a proviral role promoting VP40 assembly, maturation, and virus budding.

#### 2.2.2. SUMOylation

Ubiquitin-like modifications are very similar to Ub in terms of function and activity. Small Ubiquitin-like Modifier (SUMO) is a PTM in which a small ubiquitin-like protein is covalently attached to its target protein. Mammals encode five distinct SUMO proteins (SUMO 1–5) from which SUMO2 and SUMO3 have high identity, whereas SUMO1, SUMO4, and SUMO5 are considered paralogs, with different subcellular distributions and target specificities. SUMO peptides can form chains through their conserved lysine residues, with poly SUMO2/3 chains being the most abundant [116]. Protein SUMOylation impacts protein subcellular localization, biomolecule interactions, and consequently, signaling pathways.

Protein SUMOylation requires a distinct set of enzymes from those involved in ubiquitination but follows similar steps. SUMO priming begins with proteolytic processing by SUMO-specific proteases (SENPs), then SUMO is activated in an ATP-dependent manner by the E1 SUMO-activating enzyme (SAE1/SAE2). Subsequently, the SUMO protein is transferred to the E2 SUMO-conjugating enzyme UBC9. Finally, UBC9 transfers SUMO to the K residue of the substrate protein with the help of the SUMO E3 ligase [117]. There are several SUMO E3 ligases described to date, and interestingly, some E3 ubiquitin ligases also possess SUMOylation ligase activity [118].

##### SUMOylation of IRF7

IRF7 is a critical transcription factor in the induction of IFN-I and the antiviral innate immune response. IRF7 is also itself an ISG, which explains a second wave of IFN-I induction during infection with many viruses. Basal levels of IRF7 are usually low in many cell types, with the exception of plasmacytoid Dendritic Cells (pDCs). This allows pDCs to produce high levels of IFN-I and; therefore, are known to be important in the early IFN-I response during viral infections. Interestingly, EBOV VP35 induces IRF7 SUMOylation in pDCs by interacting with the E2 SUMO Ubc9, as well as PIAS1 and Topors E3 SUMO-ligases. Interaction of VP35, IRF7, Ubc9 and PIAS1 results in IRF7 SUMOylation and decreased activity [119]. IRF3 SUMOylation is also increased when VP35 is overexpressed, which may be a redundant mechanism by VP35 to inhibit IRF3 transcriptional activity. SUMOylation of cellular factors due to EBOV infection to inhibit antiviral immune response shows how effectively the hijacking of PTM machinery by the virus occurs in EBOV-infected cells.

##### SUMOylation and Ubiquitination Interplay in EBOV VP24

Although the SUMOylation of OEV viral proteins has not been as thoroughly studied as ubiquitination, some reports have identified viral proteins associated with SUMOylation machinery. The minor matrix protein VP24 contributes to EBOV pathogenesis by inhibiting the import of phosphorylated STAT1 into the nucleus, thereby blocking the signaling cascades of the IFN-I system [120]. VP24 was shown to interact non-covalently with SUMO through its SUMO-interacting motif (SIM), and mutation of this domain resulted in the inability to downregulate the IFN-I pathway [121]. Interestingly, this SIM domain is also required for the interaction between VP24 and the deubiquitinating enzyme ubiquitin-specific-processing peptidase 7 (USP7). The interaction between USP7 and VP24 through its SIM domain reduces VP24 ubiquitination. Furthermore, mutation of one ubiquitination site in VP24 results in a more efficient IFN-I pathway suppression, which would suggest that ubiquitination of this residue could modulate different functions of VP24 or could be a mechanism through which the cell fights off IFN-I suppression by EBOV [121]. The recruitment of USP7 through non-covalent SUMO peptides anchored to VP24, which induce VP24 de-ubiquitination, highlights a complex interplay between these PTMs that goes beyond covalent peptide conjugation and opens the possibility for new treatment development.

##### SUMOylation and EBOV VP40

In addition to the role of VP40 ubiquitination in budding, its SUMOylation has also been studied through in vitro and cell culture assays. VP40 undergoes SUMO1 and SUMO2 modifications, the latter one through its K326 residue [122]. VP40 SUMOylation results in greater stability of the viral protein when protein production is inhibited, and SUMOylated VP40 was detected in VLPs, which would suggest that this modification also plays a role in EBOV VLP production. Similar to VP24, SUMOylation of VP40 is related to ubiquitination. Inhibition of VP40 SUMO results in increased ubiquitination and may affect protein half-life through proteosome degradation. The interplay between these two PTMs needs to be further studied to better understand the anti- or proviral roles of each lysine residue that undergoes either SUMOylation, ubiquitination, or both.

### 2.3. Glycosylation

Glycosylation of secreted proteins is one of the most frequent and abundant PTMs, significantly affecting their solubility, folding, assembly, sorting, trafficking, and stability. N-linked glycosylation occurs on asparagine residues of Asn-X-ser/Thr sequons (where X is any amino acid except proline) [123]. This process occurs co-translationally early in the endoplasmic reticulum (ER). Subsequently, other enzymes trim down and extend the glycans, resulting in different classes of N-glycans, such as oligomannose, hybrid, and complex-type structures. In the mucin-type O-linked glycosylation, serine, threonine, and tyrosine residues are covalently linked to a GalNac residue by polypeptide N-acetylgalactosaminetransferases (ppGalNAcT). This modification is added as the protein passes through the Golgi apparatus [124]. Although no specific amino acid sequon has been identified for O-linked glycosylation, this modification can be found in mucin-like domains with high levels of serine, threonine, and proline residues. Glycosylation of viral envelope proteins is known to be an important mechanism of immune evasion, by masking epitopes that can potentially be recognized by antibodies [125]. In addition, envelope viral proteins can use glycosylation to promote attachment to cellular receptors [126]. Therefore, glycosylation plays critical roles during viral infections, with significant implications in tropism and the development of therapeutic strategies.

#### Glycosylation of GP

The transmembrane OEV GP is a heavily glycosylated viral protein that mediates receptor binding and membrane fusion. In addition to their contribution to the viral entry, GP itself exhibits direct pathogenicity to host cells. The overexpression of GP in cell culture can induce cell rounding and detachment. Interestingly, these effects are dependent on the presence of the extracellular mucin-like domain (MDL) of GP [105], that contains predicted sites of the type O-glycosylation [127]. The cell deadhesion effect of GP depends on the enzyme ppGalNAc-T1; the absence of this enzyme results in a reduction in the O-glycans of GP and loss of the pathogenic effect [128].

Glycosylation can also affect GP processing. There are up to seventeen N-linked glycosylation sites in GP, fifteen of which are in the GP_1_ subunit, while the GP_2_ subunit contains only two. In GP_1_, these sites are responsible for receptor binding, whereas the sites present in the membrane fusion subunit GP_2_ play an important role in GP expression, stability, and cell entry, and are conserved across all mammalian filoviruses [129,130]. Although the N-glycan structures are conserved between different Ebola viruses, the O-glycan structures vary widely between strains [127].

The use of high-throughput techniques has allowed the identification of new potential glycosylation sites in the GP protein. Approximately fifty different predicted sites for N-glycan structures are present in GP_1,2_, including high-mannose, hybrid, and bi-, tri-, and tetra-antennary complex glycans, with or without fucose and sialic acid [127]. Due to their roles in viral entry, pathogenesis, and as a molecular shield that protects the viral particle from recognition by neutralizing antibodies, and as a modulator of proteolytic processing by cellular caspases [131], a detailed characterization of the different patterns of glycosylation on GP is needed to better understand the biology and pathogenesis of OEV.

### 2.4. Protein Acetylation

Acetylation is a PTM in which a protein is linked to an acetyl group by an acetyl-transferase enzyme. Protein acetylation can occur in the N-terminus of the target protein (N-acetylation), in lysine residues (KAc), or in serine or threonine residues (O-Acetylation), each modification catalyzed by distinct enzymes and results in different effects on the target protein [132].

Protein acetylation impacts the structure of the modified protein and, thus, its activity and association with other biomolecules. This PTM has been extensively studied in histones, where acetylation of certain lysine residues can facilitate the unwinding of DNA around the histone and facilitate gene transcription. Recently, the importance of acetylation of proteins other than histones has been identified to regulate several cellular processes, such as cell cycle progression, stress response, and the antiviral immune response [133].

#### 2.4.1. Acetylation in OEV

Viral protein acetylation has been characterized for several viruses, often associated with the modulation of transcription or replication of viral genome activity, modulating the formation of ribonucleoprotein complexes. In filoviruses, the KAc of VP40 and NP from EBOV by histone acetyltransferases has been determined through in vitro assays and mass spectrometry. Identification of candidate lysine residues for KAc in NP [134] are present in its VP35-binding domain or RNA-binding cleft, which would suggest that this PTM is involved in the formation of the VP35-NP complex or NP association with the viral genome. VP40 acetylation candidate lysine residues occur in its basic patch motif, important for multimer formation, which would suggest that KAc is involved in homo-oligomer formation of VP40. These results suggest that viral protein acetylation during OEV infection may play a major role in modulating viral protein function, allowing for the virus to successfully replicate in the host cell.

##### NEDD4 Acetylation by P300

The previously discussed E3-Ub ligase NEDD4, which induces VP40 ubiquitination for oligomer formation and incorporation into virions, modulates its catalytic activity through acetylation by the acetyltransferase P300. Overexpression of P300 results in a VP40 increased ubiquitination, and virion production in P300-knockout cells is reduced when infected with EBOV. These results show a complex interplay between different PTMs, such as acetylation and ubiquitination, to drive a successful viral replication and provide new novel targets for virus control [135].

### 2.5. Acylation

Protein acylation is an important PTM that consists of the addition of acyl groups to specific residues of a target protein, such as palmitic or myristic acids [132], often to provide a lipid anchor that associates the protein with a lipid membrane if the unmodified protein is highly soluble.

Protein acylation is involved in multiple cellular processes and pathways in both homeostasis and stress conditions as a critical regulator of protein sub-cellular locations under different stimuli and protein–protein interactions [132]. Moreover, protein acylation alteration in infected cells from different viruses has been vastly studied, with both cellular and viral proteins undergoing this PTM with proviral or antiviral consequences [136].

S-palmitoylation is the reversible addition of a palmitic acid molecule to cysteine residues, usually due to a palmitoyl transferase enzyme. Palmitoylation of viral proteins by the host acylation machinery has been demonstrated for selected viruses, with important implications in their function, stability, and subcellular location [61,137].

#### OEV Protein Acylation

It has been reported that the Marburg virus GP protein is myristoylated and palmitoylated. Point mutation experiments determined that cysteine residues 670 and 672, which are highly conserved among filoviruses, are S-palmitoylated. However, in an EBOV pseudorivus that contains both mutated cysteines and no detectable palmitoylation, there is no effect on GP function regarding entry [138,139]. Further studies need to be performed to determine the role of palmitoylation and other acylation PTMs in OEV viral proteins.

## 3. Future Directions

Post-translational modifications have long been known to have regulatory cellular functions, and many studies have demonstrated how viruses can manipulate host PTM enzymes to enhance virus replication and/or prevent their role in antiviral immunity. It is only recently that the role of PTMs on viral proteins, other than glycosylation (e.g., ubiquitination), has been more appreciated. Importantly, increasing evidence indicates that ubiquitination can be present in infectious particles, adding complexity to the function of ubiquitinated viral proteins. Since ubiquitination generally occurs at low frequency, it has been challenging to perform structural studies on ubiquitinated proteins. Furthermore, the presence of a mixed population of viral particles containing ubiquitinated and non-ubiquitinated viral proteins suggests that, within a virus population, infectivity could be defined by quasispecies (different mutations in viral genomes from different viral particles), but also by mixed levels or abundance of a specific PTM. To add complexity, free ubiquitin has also been shown to be present in viral particles [94,95,96], including in EBOV [93]. Most importantly, future studies need to focus on primary cells and cell types that are naturally targets of specific viruses, because PTMs and PTM enzymes are expressed in a cell type-specific manner, and PTMs could provide cell tropism [140]. This could also affect species tropism, as it is currently believed that the natural host of EBOV may be a bat species, in which the conservation of PTM enzymes has not been studied. One of the challenges is working with highly pathogenic EBOVs, which need to be handled in Biosafety level 4 facilities (BSL4). While the trVLP system can be useful for some mechanistic studies, innovative systems that do not require transfection need to be developed.

Although phosphorylation, glycosylation, ubiquitination, and lastly SUMOylation have been the most commonly studied, proteomic analysis assisted by high-resolution experimental techniques has allowed the discovery of less abundant PTMs, including acetylation, prenylation, palmitoylation, myristoylation, ADP-ribosylation, NEDDylation, and ISGylation. Since some of these PTMs may be involved in the regulation of the immune system, it is important to investigate how viruses, including EBOV, can modulate the interaction between these PTMs and their target proteins, and whether viral proteins themselves are modified by some of them. Furthermore, computational approaches and docking simulations have emerged as novel tools that allow the prediction of sites in cellular or viral proteins that can potentially be modified by PTMs. Additionally, through this technique, it is also possible to identify molecules that can disrupt these interactions, making it a valuable tool for the discovery and development of potential antiviral drugs. Finally, the development of novel and more sensitive techniques to identify PTMs, other than Mass Spectrometry, will help advance the field.

## Figures and Tables

**Figure 1 viruses-17-01640-f001:**
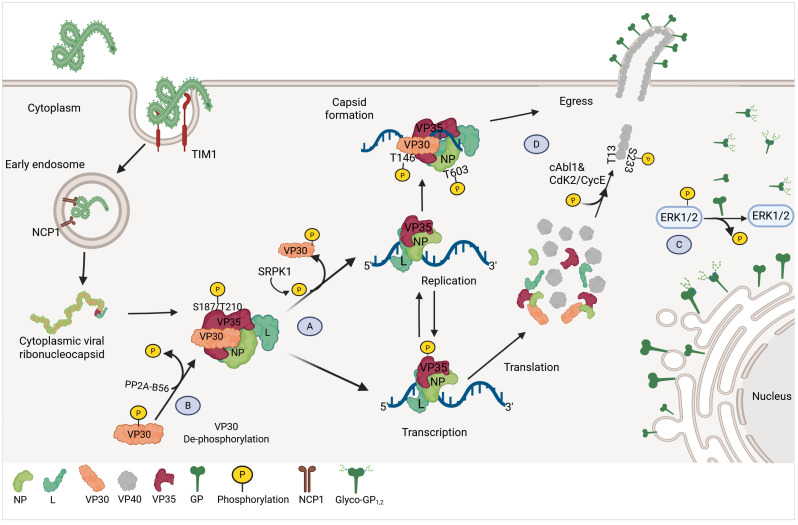
Relevance of phosphorylation during Ebola virus replication. Once the virus enters the cell and the cytoplasmatic viral ribonucleocapsid is released into the host cytoplasm, cellular kinases and phosphatases can regulate the activity and interactions of viral proteins. (**A**) Dephosphorylation of VP30 by the cellular phosphatases PP2A and B56 decreases the replication activity of the replication complex and increases the transcriptional activity. (**B**) VP35 is phosphorylated at S187 and T210, favoring its interaction with NP and enhancing viral RNA transcription. (**C**) GP can reduce the phosphorylation of ERK1/2, enhancing GP-induced toxicity. (**D**) During viral nucleocapsid assembly and egress, VP40 is phosphorylated at S233-T13, while VP30 and NP are phosphorylated at positions T146 and T603, respectively. Created in BioRender.com.

**Figure 2 viruses-17-01640-f002:**
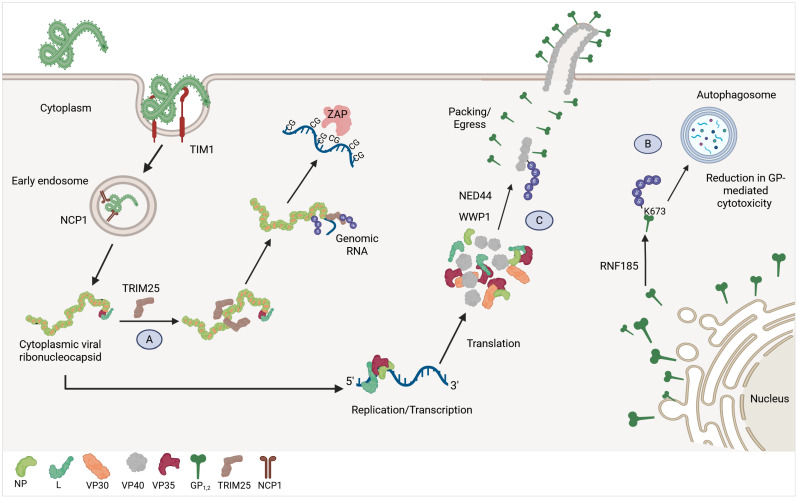
The role of ubiquitination during Ebola virus infection. Ebola virus enters the cell through its interaction with TIM-1 at the plasma membrane and NCP1 in the endosome. After entry, the viral ribonucleocapsid is released into the cytoplasm (**A**) TRIM25 induces NP ubiquitination, promoting NP dissociation from viral RNA. Then, RNA becomes accessible to the antiviral protein ZAP, which binds to the viral RNA and targets it for degradation. (**B**) The GP protein is polyubiquitinated by the E3-Ub ligase RNF185, facilitating its autophagy-mediated degradation. (**C**) VP40 is ubiquitinated by E3-Ub ligases NED44 and WWP1, favoring its oligomerization. Created in BioRender.com.

**Figure 3 viruses-17-01640-f003:**
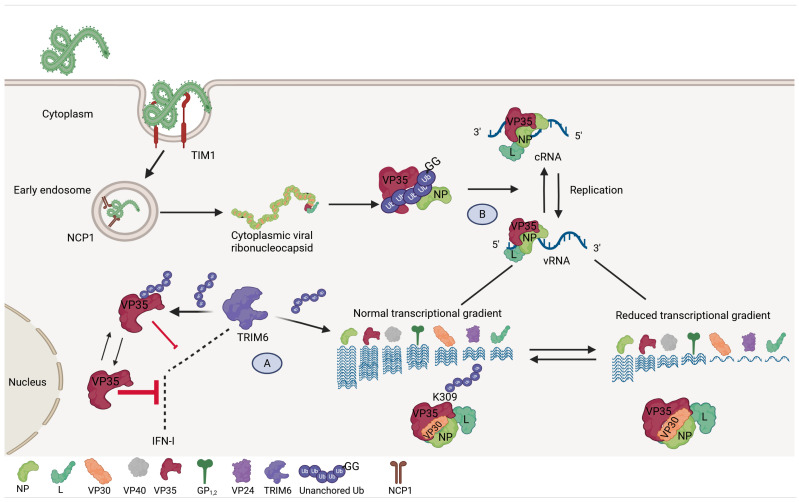
Ubiquitination of VP35 can modulate EBOV replication. (**A**) Covalent ubiquitination of VP35 on K309 by TRIM6 promotes its interaction with the viral polymerase L, leading to viral transcription and preventing premature encapsidation. Furthermore, VP35 ubiquitination reduces IFN-I inhibition. (**B**) Unanchored K63-linked polyubiquitin chains interact non-covalently with VP35, promoting its interaction with NP and enhancing polymerase functions. Created in BioRender.com.

**Table 1 viruses-17-01640-t001:** Ebolavirus outbreaks.

Virus	Period (Year)	Country	Human Confirmed Cases	Case Fatality Rate
SUDV	1976	South Sudan	284	53%
EBOV	1976	DRC	318	88%
EBOV	1977	DRC	1	100%
SUDV	1979	South Sudan	34	64%
EBOV	1994	Gabon	52	52%
TAFV	1994	Cote d’Ivoire	1	0%
EBOV	1995	DRC	315	81%
EBOV	1996	Gabon	31	67%
EBOV	1996	Gabon	60	75%
SUDV	2000–2001	Uganda	425	53%
EBOV	2001–2002	Gabon	65	80%
EBOV	2001	ROC	59	75%
EBOV	2003	ROC	143	90%
EBOV	2003	ROC	35	83%
SUDV	2004	South Sudan	17	41%
EBOV	2005	ROC	12	83%
EBOV	2007	DRC	264	71%
BDBV	2007	Uganda	149	25%
EBOV	2008–2009	DRC	32	47%
SUDV	2011	Uganda	1	100%
BDBV	2012	DRC	62	55%
SUDV	2012	Uganda	24	71%
SUDV	2012	Uganda	7	57%
EBOV	2013–2016	Guinea	28,656	40%
EBOV	2014	DRC	69	71%
EBOV	2017	DRC	8	50%
EBOV	2018	DRC	54	61%
EBOV	2018–2020	DRC	3470	66%
EBOV	2020	DRC	130	42%
EBOV	2021	DRC	12	50%
EBOV	2021	Guinea	23	52%
EBOV	2021	DRC	11	55%
EBOV	2022	DRC	5	100%
EBOV	2022	DRC	1	100%
SUDV	2022	Uganda	164	47%
SUDV	2025	Uganda	12	40%
EBOV	2025	DRC	42 *	65%

Based on [11] and https://www.cdc.gov/ebola/outbreaks/index.html (accessed on 10 February 2025). DRC: Democratic Republic of the Congo; ROC: Republic of Congo. EBOV: Ebola virus; SUDV: Sudan virus; BDBV: Bundibugyo virus; TAFV: Taï Forest virus. * Data reported as of 10 February 2025.

## Data Availability

No new data were created or analyzed in this study.

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
