# Peer review of "The Role of Posttranslational Modifications During Ebola Virus Infection"

_viruses, 2025, doi:10.3390/v17121640_

Round 1
Reviewer 1 Report
Comments and Suggestions for Authors
The authors are to be commended for a thorough and well written review of the functional molecular biology of posttranslational modifications of viral and host proteins during infection by orthoebolaviruses. Note that I did not capitalize or italicize "orthoebolaviruses", as this is referring to these viruses as a group, and the current ICTV about nomenclature require virus names to be uncapitalized in standard font, as opposed to species and genera names, which are capitalized and italicized.
So, I have no issues with the scientific review content, which is well done. My chief recommendation for revision is to conduct a thorough examination of the way in which names and abbreviations are used throughout the manuscript, to conform with current standards.
An up to date and authoritative discussion of the taxonomy and nomenclature of viruses in this class is found in this 2023 article:
doi: 10.1007/s00705-023-05834-2
More specifically, the authors should consult the two tables therein, which are found here:
https://link.springer.com/article/10.1007/s00705-023-05834-2/tables/1
https://link.springer.com/article/10.1007/s00705-023-05834-2/tables/2
Table 2 enumerates common nomenclature problems in manuscripts on the virology of filoviruses. I suggest that the authors pay attention to any relevant sections of the table and edit their manuscript accordingly.
The evolution of nomenclature in this field is in some ways counterintuitive and vexing to deal with. For example, the construct "ebolavirus" as a standalone single word is no longer permitted or correct except as a component of "orthoebolavirus" which may also be "Orthoebolavirus" when occurring in genus or species names. Exceptions would be quotes and titles of previously published articles, commercial products, etc.
Another problem is in the use of the abbreviation EBOV throughout the manuscript. If the authors intend to solely be discussing results in relation to the virus formerly known as Zaire ebolavirus, now simply Zaire virus (see table 1 cited above), then use of EBOV is appropriate. If they're trying to talk about the class of orthoebolaviruses, which seems appropriate to me, as outbreaks of e.g, Sudan virus are associated with similar pathogenic mechanisms, then the authors are faced with the limitation that there is no accepted abbreviation for orthoebolavirus or orthoebolaviruses, when talking about the collective group of these viruses, nor is there even an accepted abbreviation for the genus Orthoebolavirus. (See my final comment below).
So here are some specific examples of how the manuscript needs to be modified, starting with the title:
EITHER
The Role of Posttranslational Modifications During Ebola virus Infection
(if the review is largely limited to studies of the Zaire orthoebolavirus, which is named "Ebola virus" with a lower case v).
OR, if the review is intended to cover orthoebolaviruses as a class:
The Role of Posttranslational Modifications During orthoebolavirus Infection
(NOT the genus name Orthoebolavirus - see bottom lines of table 2 cited above).
It seems that it would be legitimate in most of the manuscript to be talking about Ebola virus (EBOV) specifically, rather than the class of orthoebolaviruses, in which case the authors just need to change Ebolavirus to Ebola virus, and continue to use the EBOV abbreviation. If they really want to be talking about the similarities of these mechanisms throughout this genus, my recommendation is starting, with the first sentence of the abstract to change it to "Orthoebolaviruses are highly pathogenic viruses...”
At the first citation of EVD, it should probably now be listed as Ebola virus disease (not Ebolavirus disease) , avoiding the now-prohibited ebolavirus. Still, this change has the unfortunate implication that EVD is specific to the Zaire virus. Thus, one could argue it should now be "orthoebolavirus disease", OVD or OEVD, but of course there are years of literature discussing EVD, so this is an unfortunate conundrum that arises with changes in nomenclature.
The body of table 1 is good, as they have used the correct abbreviations for the viruses, but in the legend to the table they have inappropriately italicized most of the virus names, which also need to be corrected by changing ebolavirus to virus, e.g., Reston virus.
Then in the first paragraph after Table 1 (bottom of p. 2), they list the viral species in this genus using outdated species names which should be updated with those in the Table I cited above, e.g. as Orthoebolavirus zairense, etc.
More critically, it is inappropriate to use the abbreviations for the virus names that were used in the body of their table 1 as abbreviations following the species names. The abbreviations for the species names of these viruses use the format ZEBOV, REBOV, SEBOV, etc.
I will leave it to the authors to apply the rules in table 2 that I cited above to correct their manuscript. Bear in mind that it is important to realize that species name should not be routinely cited in virological work, as explained at the bottom of that table. As the results being reviewed pertain to viral entities, the virus names should be applied. But if you are talking about these viruses as a group, you're pretty much restricted to just spelling out "orthoebolaviruses" in full. If it was my paper, I’d be tempted to introduce an abbreviation like OEV to refer to orthoebolaviruses, then use that in place of EBOV throughout the manuscript. It seems to me such an abbreviation is badly needed…
Hopefully this is helpful and not too confusing.
Reviewer 2 Report
Comments and Suggestions for Authors
Moreno-Contreras and colleagues present a thorough review of the role of the most common post-translational modifications during EBOV infection. This is a crucial area of study as it informs on virus-host interactions, physical and functional protein modifications and their effects on the virus lifecycle, and possible strategies for antiviral development. Overall this is a timely article that is well written and nicely organized. Although it is not possible to include every key reference in a review article, I would encourage the authors to review the most recent literature in the last 2-3 years, as some relevant papers on this topic have not been cited. While I have no major concerns, I would suggest that the authors consider simplifying the two figures, as they are quite complex and tedious to the eye. Perhaps each PTM and its effect(s) on specific stages of EBOV replication could be represented in separate figures?
Round 2
Reviewer 2 Report
Comments and Suggestions for Authors
The authors addressed my concerns adequately.